# Path Planning of Pattern Transfer Based on Dual-Operator and a Dual-Population Ant Colony Algorithm for Digital Mask Projection Lithography

**DOI:** 10.3390/e22030295

**Published:** 2020-03-03

**Authors:** Yingzhi Wang, Tailin Han, Xu Jiang, Yuhan Yan, Hong Liu

**Affiliations:** 1School of Electronics and Information Engineering, Changchun University of Science and Technology, Changchun 130022, China; wyz@cust.edu.cn (Y.W.); jiang273944219@163.com (X.J.); yanyuhan0304@163.com (Y.Y.); 2School of Opto-electronic Engineering, Changchun University of Science and Technology, Changchun 130022, China; liuh19694@163.com

**Keywords:** path planning, dual-operator and dual-population ant colony algorithm, adaptive algorithm, pattern transfer, DMD

## Abstract

In the process of digital micromirror device (DMD) digital mask projection lithography, the lithography efficiency will be enhanced greatly by path planning of pattern transfer. This paper proposes a new dual operator and dual population ant colony (DODPACO) algorithm. Firstly, load operators and feedback operators are used to update the local and global pheromones in the white ant colony, and the feedback operator is used in the yellow ant colony. The concept of information entropy is used to regulate the number of yellow and white ant colonies adaptively. Secondly, take eight groups of large-scale data in TSPLIB as examples to compare with two classical ACO and six improved ACO algorithms; the results show that the DODPACO algorithm is superior in solving large-scale events in terms of solution quality and convergence speed. Thirdly, take PCB production as an example to verify the time saved after path planning; the DODPACO algorithm is used for path planning, which saves 34.3% of time compared with no path planning, and is about 1% shorter than the suboptimal algorithm. The DODPACO algorithm is applicable to the path planning of pattern transfer in DMD digital mask projection lithography and other digital mask lithography.

## 1. Introduction

Lithography plays a leading role in micro-nano manufacturing and many other industrial applications. The traditional lithography technology is to convert the light to the substrate through the physical mask. After converting, the photoresist on the substrate is exposed to complete the pattern transfer. Then, lithography was achieved through the subsequent process. However, the high cost, time consuming nature, and lack of flexibility of mask manufacturing have become the bottlenecks of lithography [1]. In recent years, electron beam [2], laser direct writing [3], laser interference lithography [4], focused ion beam [5], and DMD projection lithography [6] are employed to solve this problem. These lithographic methods are mostly single-spot exposure, and even the area array multi-point projection exposure of DMD has a small exposure area at high resolution. Compared with the physical mask, digital mask lithography has a lower pattern transfer efficiency. When performing micron lithography on large-area patterns, the principle of projection lithography system based on DMD [7] is shown in Figure 1a. The system is composed of light source, DMD, optical system (Fourier lens, Fourier filter, reduction lens), software system, 3D mobile platform, etc. In the process of digital mask exposure, DMD has the advantages of large exposure area, high resolution, high production efficiency, and low cost [8,9,10], which is widely used in MEMS production [11], micro optical device processing [12,13,14], 3D micro-nano structure processing [15,16,17], pattern transfer of printed circuit board (PCB) [18], etc. The output beam from the light source is irradiated onto the DMD after passing through the reflector. According to the designed digital pattern, the computer controls the DMD micromirror to modulate the light to generate a digital mask. After the modulated beam is reduced by the optical system, the photoresist on the substrate is exposed to realize the pattern transfer. For some large-area pattern transfer applications, such as transferring conductive patterns on a PCB [18], the DMD projection area is much smaller than the substrate area. The substrate is divided into multiple cells by the software, each cell representing the area where the DMD is projected once, as shown in Figure 1b. The conductive pattern of the PCB does not completely cover the substrate. Green represents cells with conductive patterns, and white represents cells without conductive patterns. When the DMD is projected to a certain cell and a pattern transfer is completed, the 3D mobile platform moves to the next cell for the next pattern transfer. If DMD projection exposure traverses every cell to transfer patterns, the exposure staying in the white cells will greatly reduce the DMD pattern transfer efficiency. If we use some algorithms to plan the walking path of the 3D mobile platform and make it move as shown by the arrow in Figure 1b, the walking distance of 3D mobile platform will be greatly shortened and the efficiency of pattern transfer will be effectively improved. In the literature [1,13,19,20,21,22], multi-DMD exposure head, exposure dose enhancement, light source uniform, Wobulation technology, and other technologies were used to improve the exposure efficiency and resolution. However, the production efficiency was reduced without the optimization of invalid stay and exposure at the nonconductive part of the transfer pattern. Figure 1b shows an enlarged image of a substrate by path planning after pattern transfer. Move the 3D platform so that the DMD is projected onto each green cell. The 3D mobile platform is required to have the shortest moving path, and the DMD cannot repeatedly project a certain green cell—for example, the motion path shown by the arrow in Figure 1b, and this path planning is similar to the traveling salesman problem (TSP) [23]. Yang et al. [24] used a genetic algorithm to plan the path of small area silicon wafer pattern transfer lithography, and the exposure efficiency of the lithography machine to the silicon wafers was increased by about 6%. However, this method does not provide a solution to the problem of large-area and no patterns to be transferred on the substrate. When solving large-scale TSP problems, genetic algorithm have the disadvantage of slow convergence, and ant colony algorithm have the advantage of solving large-scale problems [25,26].

In 1991, Italian researcher Marco Dorigo et al. [27,28] proposed a highly innovative MetaHeuristic Algorithm inspired by the real foraging behavior of ants: Ant System. After decades of development, researchers have proposed an ant colony optimization (ACO) algorithm [29] and ant colony system (ACS) algorithm [30]; good results were obtained. However, there are still problems of slow convergence and poor quality of the solution when dealing with large-scale problems. The conventional optimization strategy of the ant colony algorithm is to improve the pheromone updating strategy, local search, and parameter selection to achieve an ideal solution or convergence rate [31]. Deng et al. [32] improved ant colony optimization algorithm from pheromone updating strategy and pheromone diffusion mechanism to solve the Scheduling Problem. Zhang et al. [33] proposed a multi-population ant colony optimization algorithm based on congestion factor and co-evolution mechanism to solve a large-scale traveling salesmen problem with better performance. Chen et al. [34] proposed an entropy-based dynamic heterogeneous ant colony optimization to solve a large-scale traveling salesman problem. Jia et al. [35] applied a local optimization strategy to optimize the ant colony algorithm, solving the problems of manufacturing time and energy consumption. Sun et al. [36] used multiple ant colonies for the solution and determined strategies for information exchange among ant colonies according to the information entropy of each population to guarantee the balance of its convergence and diversity. Guan et al. [37] used information entropy to choose the positive or negative feedback strategy. A repulsive operator was used in literature [38] used to improve the ant colony algorithm. Li et al. [39] improved the ant colony algorithm by using crossover operator to enhance the global search ability. Mohsen et al. [40] improved the ant colony algorithm by using a mutation operator to increase the ants’ population diversity.

It can be concluded from the above literature that local pheromone update strategy or global pheromone update strategy is very effective to improve the ant colony algorithm. However, there still exists the disadvantage of slow convergence speed or easily falling into local optimum values to optimize large-scale problems. Inspired by the literature [36,37,38,39,40], various types of operators are beneficial to optimize pheromone update strategy, and information entropy can classify the population. Therefore, this paper proposes a new dual operator and dual population ant colony (DODPACO) algorithm to achieve the goals of accelerating convergence and improving the quality of the solution. Using the DODPACO algorithm to plan the walking path of DMD 3D mobile platform based on the stepper system DMD pattern transfer system can effectively reduce the walking path length of the DMD 3D mobile platform to improve the exposure efficiency in the process of PCB pattern transfer.

## 2. Methods

### 2.1. ACS Algorithm

In the ACS algorithm, the state transition rule is as follows: $\tau_{mn}$ represents the pheromone concentration between city *m* and city *n*; $\eta_{mn}$ represents the heuristic information on the edge (m,n), and is the reciprocal of the distance between city *m* and city *n*. An ant *k* located in city m selects the next city *n* according to Equation (1) [29]. $allowed_{k}$ represents the collection of cities that ant *k* can choose. α reflects the influence of pheromone concentration on the path selection of subsequent ants, which is called the pheromone factor. β reflects the influence of distance length between cities on ant path selection, which is called expectation heuristic information factor:(1)$$P_{mn}=\{\begin{array}{cc} \frac{{\left[\tau_{mn}\right]}^{\alpha }{\left[\eta_{mn}\right]}^{\beta }}{\sum_{u\in allowed_{k}}{\left[\tau_{mu}\right]}^{\alpha }{\left[\eta_{mu}\right]}^{\beta }} & n\in allowed_{k}\\ 0 & else \end{array}$$

In ACS, only globally optimal ants are allowed to release pheromones, which is called global pheromone update strategy. The global pheromone update rule is given by Equation (2) [30]:(2)$$\tau_{mn}\leftarrow \{\begin{array}{cc} \theta \times \tau_{mn}+\left(1-\theta \right)\times {\left(L_{gb}\right)}^{-1} & \left(m,n\right)\in global\;optimal\;solution\\ \theta \times \tau_{mn} & else \end{array}$$
where $L_{gb}$ is the global shortest path length. The parameter θ(0<θ<1) represents the pheromone retention parameter.

In addition to the global pheromone update rule, ACS also contains a local pheromone update rule. Each time an ant passes through an edge (m,n) during an iteration to update the pheromone concentration on the path by calling Equation (3). ε represents the pheromone retention parameter, which satisfies 0<ε<1:(3)$$\tau_{mn}\leftarrow \varepsilon \times \tau_{mn}+\left(1-\varepsilon \right)\times \tau_{0}$$

### 2.2. Improvement Strategy

#### 2.2.1. Self-Adaptive Ant Colony Division

The ant colony is divided into two populations: yellow ant colony and white ant colony. Different operators are set for the two populations, and then the relationship between the two populations is adjusted dynamically by adjusting the number of ants of the two populations adaptively, so as to solve the problem of contradiction between the quality of solution and the speed of convergence. The yellow ant colony plays a major role in the early stage to improve the quality of the solution, whereas the white ant colony plays a major role in the late stage to accelerate the convergence speed.

Assuming that the total number of the ants is *G*, $G_{w}$ represents the number of white ants in the white ant colony, $G_{y}$ represents the number of yellow ants in the yellow ant colony, *k* represents the current number of iterations, and *K* represents the maximum number of iterations. According to Equation (4), the number of ants in the yellow and white ant colonies is obtained:(4)$$\{\begin{array}{c} G_{w}=\frac{1}{4-3\frac{k}{K}}G\\ G_{y}=G-G_{w} \end{array}$$

In the initial implementation of the algorithm, the number of white ant colonies and yellow ant colonies is about 25% and 75% of the total ant colonies, respectively. When iterating to about 2000 generations, the number of white ant colonies is about 50% of the total ant colonies. When all the iterations are completed, all ants belong to the white ant colony. The white ant colony has a low proportion in the early stage and plays a small role, whereas the number in the later stage increases rapidly and plays a major role in improving the convergence rate of the algorithm in the later stage. Correspondingly, the yellow ant colony plays a great role in the early stage to improve the quality of the solution. With the rapid decrease of the number in the later stage, its influence on the algorithm gradually decreases.

There are *x* paths after *t* iterations. The ratio of the ants on each path is $P_{m}=a_{m}/G\;\left(m=1,2,\ldots ,x\right)$, $a_{m}$ is the number of ants on path *m*, and the total number of ants that have participated in the iterations is A, and $A=\sum_{m=1}^{x}a_{m}$, A≤G. The information entropy expression for defining the path diversity is Equation (5):(5)$$e\left(t\right)=-c\sum_{m=1}^{x}P_{m}\left(t\right)\log_{2}P_{m}\left(t\right)$$

Formula (5) represents the uncertainty of ant selection path in the *t*-th iteration, where *c* is the weight coefficient. When *G* ants generate *G* paths iteratively, the number of generated paths is the largest, and the information entropy is the largest. With the path optimization, ants gradually gather on the optimal paths, and the information entropy becomes smaller. Therefore, the change of information entropy can reflect the evolution degree of ant population.

When the ant colony evolves to a certain degree, the convergence of the algorithm will be accelerated. To describe this degree, the proportion of ants choosing the same path to the total number of ants is ρ; then, Equation (6) represents the value of information entropy at this point:(6)$$W=-c\left[\rho \log_{2}\rho +\left(1-\rho \right)\log_{2}\left(1-\rho \right)\right]$$

When ρ×G ants choose the same path, which is (ρ)≥e(t), the algorithm will enter a new stage. In this stage, the yellow ant colony is killed, and the corresponding number of white ants is activated to accelerate the convergence of the algorithm.

#### 2.2.2. Feedback Operator

All ants complete one iteration and record the path length of each ant in this iteration. If the iteration finds a better solution, the pheromone distribution of the current path is beneficial to optimization, which can accelerate the accumulation of pheromones and speed up the acquisition of the optimal solution; if the result of the iteration is equal to the current optimal solution, it is necessary to continue the optimization; if the result of the iteration is inferior to the current optimal solution and the pheromone distribution of the current path is not conducive to the optimization, then the method of slowing down the accumulation of pheromones is used to change the choice of the path and look forward to finding the optimal solution. The shortest path of the *r*-th iteration is $\Psi_{min}$, the average path length of this iteration is $\Psi_{mean}$, and the optimal path of the previous (r−1) iterations is $\Psi_{opt}$. Thus, Equation (7) is the feedback operator formula obtained:(7)$$\delta =1-cos\left(\frac{\Psi_{mean}-\Psi_{min}-1}{\Psi_{mean}-\Psi_{opt}-1}\times \frac{\pi }{2}\right)$$

When the shortest path of the *r*-th iteration is less than the optimal path of the previous (r−1) iterations, 1<δ≤2, a shorter path is found. The feedback operator plays a positive feedback role in pheromone update and path selection of the next iteration. If the shortest path of the *r*-th iteration is greater than or equal to the optimal path of the previous (r−1) iterations, 0≤δ≤1. If no shorter path is found, no function or negative feedback function will be played.

#### 2.2.3. Load Operator

When the number of cities is b and the ant is located in city *m*, a certain amount of pheromones are accumulated on the optional (b−1) roads. Load operator is introduced to measure the load degree of pheromone on a certain path. Load operator represents the ratio of pheromone concentration on a path to the sum of pheromone concentration on (b−1) paths, as shown in Equation (8):(8)$$\omega_{mn}=\frac{\tau_{mn}\left(t\right)}{\sum_{r=1}^{b-1}\tau_{mn}\left(t\right)}$$

We can conclude that, when *b* is large enough, the $\omega_{mn}$ value is close to 0. The upper limit λ of the load operator is set, and when the load operator on the path (m,n) exceeds λ, the concentration of the path information is regulated by reducing the information on the path. The function of load operator: avoid falling into local optimization in the early stage, change its upper limit to make it fail properly in the later stage, and reduce its influence on convergence speed.

### 2.3. Self-Adaptive Dual-Population Ant Colony

#### 2.3.1. Convergence Rate Optimization

In the classical ant colony algorithm, the conventional method is to update pheromones locally or globally. This single pheromone update method will reduce the convergence speed, and it is easy to fall into local optimum, and it is difficult for it to jump out. In the improved algorithm, the solution results of a single white ant can be locally updated, and the solution results of the white ant colony can be globally updated by a comprehensive application. The two pheromone update rules of the improved algorithm are shown in Equation (9), where ω is the load operator of side (m,n), δ is the feedback operator, ε is the pheromone retention parameter during local update, and θ is the pheromone holding parameter during a global update:(9)$$\tau_{mn}\left(t+1\right)=\{\begin{array}{cc} \Upsilon \times \tau_{mn}\left(t\right)+\omega_{mn}\times \delta \times \left(1-\Upsilon \right)\times \tau_{0} & \left(\omega_{mn}\geq \lambda \right)\cap \left(\Upsilon =\varepsilon \cup \Upsilon =\theta \right)\\ \Upsilon \times \tau_{mn}\left(t\right)+\left(1+\omega_{mn}\right)\times \delta \times \left(1-\Upsilon \right)\times \tau_{0} & else\cap \left(\Upsilon =\varepsilon \cup \Upsilon =\theta \right) \end{array}$$

In the early stage of the algorithm, the next node is selected through the distance between nodes, and the nodes are selected many times, resulting in excessive accumulation of pheromone concentration on the path. Under the positive feedback mechanism, pheromone accumulation continued, which affects the selection of the next iteration time point, so that it is impossible to jump out of the local optimal to obtain a better solution. When $\omega_{mn}\geq \lambda$, as $\omega_{mn}$ is small, the accumulation of pheromones on this edge can be limited, and the excess of pheromones on a single edge can be rejected. According to the definition of δ, if the optimal path is found after the (*t* + 1)-th iteration, the pheromone accumulation can be accelerated by Equation (9) to achieve positive feedback. If no better path is found, the accumulation of pheromone is restricted by Equation (9) to realize negative feedback. At the end of iterations, upper limits $\lambda_{1}$ and $\lambda_{2}$ are adjusted to update the pheromone on the path without getting trapped in local optimization, so as to accelerate pheromone accumulation, accelerate convergence, and ensure the diversity of the improved algorithm after the yellow ant colony is killed.

In the later stage, in order to accelerate the convergence speed, kill the yellow ant colony and activate the corresponding number of white ant colony, the white ant colony path probability selection is calculated according to Equation (1).

#### 2.3.2. Optimization of Solutions

Yellow ant colony adjusts the path probability selection when the ant colony system iterates. Using the rule of pheromone accumulation in ACS algorithm, if $\phi >\phi_{0}$, then the ants select according to Equation (10):(10)$$P_{mn}^{k}\left(t\right)=\{\begin{array}{cc} \frac{\delta {\left[\tau_{mn}\left(t\right)\right]}^{\alpha }{\left[\eta_{mn}\right]}^{\beta }}{\sum_{s\in allowed_{k}}\delta {\left[\tau_{ms}\left(t\right)\right]}^{\alpha }{\left[\eta_{ms}\right]}^{\beta }} & n\in allowed_{k}\\ 0 & else \end{array}$$

Here, δ is the feedback operator. In the rules of the classical ACS algorithm, the pheromone concentration and the distance between cities are the two factors that affect the next node selection. In the early stage, the determinant of the next node selection is the distance between cities. If the pheromone concentration on some paths is too high, it will lead to a high probability of being selected in the next iteration, and the path will be limited to several cities, so it is difficult to get a better solution. The introduction of δ makes the probability selection also affected by the quality of the solution generated from the last iteration. The selected target is adjusted by positive and negative feedback in order to improve the randomness of probability selection. In the early stage of the algorithm, the probability selection is adjusted according to whether the better solution is obtained, so as to avoid the local optimization caused by too much influence on the distance between cities in the early stage. When δ>1, a better solution is obtained, which generates positive feedback on the probability selection; otherwise, negative feedback or no influence is generated. The result of the improved algorithm is that the path is fixed on multiple paths instead of one. However, it is difficult to jump out of the local optimum in the later stage when it is trapped in the local optimum. Therefore, in this paper, when the ant colony evolves to a certain degree, that is, when the information entropy reaches a certain value, kill the yellow ant colony and activate the corresponding number of white ants so as to invalidate the operator, and use the white ant colony operator to jump out of the optimum.

#### 2.3.3. Algorithm Flow

The algorithm flow is described as follows: After setting the initialization parameters, ACS is used to complete the first iteration, and its data are used to calculate the pheromone. Use Equation (5) to calculate the information entropy after each iteration. When the information entropy reaches the preset value W(ρ), the yellow ant colony will be killed and the corresponding number of white ants will be activated, so that the operator will fail, and the algorithm will enter the accelerated iteration until the preset maximum number of iterations. The function of entropy is to measure the evolution degree of the ant colony system, that is, to divide the different tasks of the algorithm in different stages, so as to achieve the balance between the solution quality and the convergence speed:

Step 1 initialize parameters such as α, β, θ, *k*, ε, ρ, $\lambda_{1}$, $\lambda_{2}$, etc.;

Step 2 use the ACS algorithm to complete an iteration;

Step 3 calculate the load operator and feedback operator;

Step 4 divide the white and yellow ant colony adaptively;

Step 5 use white ant colony and yellow ant colony operators to iterate;

Step 6 calculate the information entropy e(t) value, and the number of iterations *t* plus 1;

Step 7 determine whether the number of iterations *t* < *T* is true, output that the current path is the optimal path if it is false, and end the calculation, where *T* is the maximum number of iterations set;

Step 8 judge the relationship between entropy e(t) and W(ρ). If W(ρ)≥e(t), the yellow ant will be inactivated and the corresponding number of white ants will be copied; otherwise, step 2 will be executed;

Step 9 calculate the load operator and feedback operator;

Step 10 use the white ant operator to iterate, and the number of iterations t plus 1;

Step 11 perform step 7;

Step 12 perform step 9.

## 3. Results and Discussion

MATLAB 2013a is used in order to verify the performance of this improved algorithm. Taking eight sets of large-scale data in TSPLIB as an example, the simulation is compared with two classical ACO and six improved ACO algorithms. In actual cases, four ACO algorithms are compared with the DODPACO algorithm.

### 3.1. Algorithm Simulation and Discussion

#### 3.1.1. Parameter Setting

At the beginning of the algorithm, the same pheromone concentration on each path causes the pheromone concentration to have little effect on the path construction, but, with the accumulation of pheromones on the path after iterations, the pheromone has more and more influence on the path construction. Therefore, a large β value and a small α value are selected to make η pay a larger role in the early stage and accelerate the path construction. In the later stage, the influence of pheromones plays a major role, so the influence of η can be approximately ignored [41]. λ is used to limit the concentration of pheromone on the path. If λ is too large, the algorithm will fail, and, if λ is too small, the convergence speed of the algorithm will be affected. The experimental results show that, when ρ=0.28, the entropy W(ρ) obtained from the predetermined value of information can well divide the early and later stages of the algorithm. λ plays a better role without affecting convergence. The parameter setting in this paper is shown in Table 1, and the value of ant number *G* is shown in Table 2.

#### 3.1.2. Simulation Results and Analysis

The error rate is to measure the difference between each kind of ACO and the optimal solution of the test set; the calculation formula is shown in Equation (11), where $R_{ACO}$ is the optimal solution found by the ACO algorithm, and $R_{min}$ is the standard optimal solution of the test set. The standard deviation represents the degree of dispersion of multiple solutions of each ACO algorithm. The calculation formula is shown in Equation (12), where *N* is the number of simulations and *r* is the mean value of the solution:(11)$$\Gamma =\left(\frac{R_{ACO}}{R_{min}}-1\right)\times 100\%$$
(12)$$\sigma \left(r\right)=\sqrt{\frac{1}{N}\overset{N}{\underset{i=1}{\sum }}{\left(x_{i}-r\right)}^{2}}$$

##### Comparison with Simulation Results of Classical ACO Algorithms

In this paper, eight groups of data in the TSPLIB test set are taken as the subject for 30 experiments on DODPACO, ACS, and MMAS, 3000 iterations per round. All the results obtained by each comparison algorithm in 30 runs are used as experimental data, and the optimal value, average value, error rate, and minimum number of iterations of the solution are obtained, as shown in Table 3. The visual graphs of the data in Table 3 are shown in Figure 2, Figure 3 and Figure 4. As can be seen in Figure 2, the error rate of DODPACO is smaller than other algorithms. For the DODPACO algorithm, the value of the error rate is between 0 and 0.92, and the maximum error rate of the ACS algorithm and the MMAS algorithm is 6.58. This result shows that, compared with the ACS algorithm and the MMAS algorithm, the gap between the optimal solution of the DODPACO algorithm and the known best solution is the smallest. The average value of the DODPACO algorithm is the smallest, and the standard deviation is mostly smaller than other algorithms. It shows that the concentration of the DODPACO algorithm is the best, and the visual graph of standard deviation is shown in Figure 3. The standard deviation of ACS in kroa200 experiment is 2 smaller than the DODPACO algorithm; this difference is not obvious. Looking at all the experiments from the data, only this value is not optimal. However, in the kroa200 experiment, the results such as the convergence speed of the DODPACO algorithm are the minimum. Figure 4 shows that, in the TSP225 experiment, although DODPACO and ACS have a small difference in the number of iterations, DODPACO has advantages in terms of error rate and standard deviation when the number of iterations is less than ACS. It can be seen from the simulation results that the DODPACO algorithm can effectively balance the contradiction between the quality of the solution and the speed of convergence.

##### Comparison with Simulation Results of Other ACO Algorithms

In order to demonstrate the superiority of the proposed DODPACO algorithm, the proposed algorithm is compared with PCCACO algorithm [42], EDHACO algorithm [34], ICMPACO algorithm [32], PSO-ACO-3opt algorithm [43], HHACO algorithm [44], and CCMACO algorithm [45]. The experiment results are shown in Table 4. The average value of the DODPACO algorithm is the smallest, indicating that the concentration of the solution is better. In the pr152 experiment, the optimal solutions of the EDHACO algorithm and the DODPACO algorithm are 73,682 and 73,683, respectively. The EDHACO algorithm solutions are smaller, and the difference between them is 1. However, for the average of this group of experiments, the DODPACO algorithm is better than the EDHACO algorithm, which are 736,905 and 74,251.6, respectively. The optimal solutions for other groups of simulations are obtained by the DODPACO algorithm. The experimental results show that the DODPACO algorithm can obtain the best optimization value and the minimum average value. Compared with the other six algorithms, the DODPACO algorithm has advantages.

### 3.2. Verification Experiments and Discussion

#### 3.2.1. Path Planning Model Establishment

In the DMD pattern transfer lithography system shown in Figure 1b, the resolution of the DMD used is 1024 × 768. The size of a single vibrating mirror is about 14 μm, and the photosensitive size of the whole DMD component is about 14 mm × 10 mm [46]. In the process of PCB pattern transfer, the resolution is required to be no more than 3 μm, and then the projected light spot of DMD needs to be miniaturized with a miniaturization ratio of 5:1. The light emitted by the light source is modulated by a DMD vibrating mirror, and the exposure area after miniaturization is reduced from 14 mm × 10 mm to 3 mm × 2 mm, while each exposure pixel is also reduced from 14 μm to 3 μm, that is, the exposure resolution can be up to 3 μm. In the application of large-area exposure, Figure 5 shows an intercept part of a PCB with a size of 60 mm × 40 mm. The DMD projection size is 3 mm × 2 mm. The PCB graph is divided into several cells with a size of 3 mm × 2 mm, and the PCB with an area of 60 mm × 40 mm is divided into 20 × 20 cells. In this example, 266 of the 400 cells have conductive patterns. Cells with conductive patterns are marked with green, green marks represent cities in the TSP problem, and cells without conductive patterns are marked with white, as shown in Figure 6 after marking. The designed PCB circuit diagram is divided into 3 mm × 2 mm cells by a computer. Detect the presence or absence of conductive patterns in each cell, and mark them as green or white, respectively. At the same time, the 1024 × 768 pixels of each cell correspond to the 1024 × 768 micromirror of the DMD. The micromirror with a circuit diagram is modulated to the “on” state and can reflect light to the surface of the photoresist. The micromirror without circuit diagram is “off”, and the reflected light cannot reach the surface of the photoresist. The conventional “S” path traversal method projects and exposes cells one by one, and the calculated path length is 1238 mm. The DODPACO algorithm is used for DMD projection exposure, that is, only 266 green cells need to be traversed, and the white cells do not need DMD to stay.

#### 3.2.2. Verification Experiments

If the DMD projection exposure machine follows the normal path (press 1, 2, 3, …, 399, 400 of the order) for exposure, the total distance of the stepping process is f = 1238 mm. According to the path planning model in Section 3.2.1, the planning path for the needs in Figure 6 is established, and the four ACO algorithms EDHACO, PSO-ACO-3opt, ACS, and MMAS are applied to compare with the DODPACO algorithm and the no-path planning method. During the simulation in the MATLAB environment, the 3000 iterations are set, and each algorithm is simulated for 10 times. The optimal path length is shown in Table 5. In the verification experiment, there are two key parameters of the 3D mobile platform. One is the shortest time from one projection position to the next, which is 0.5 s; the other is the single DMD exposure time which is 2.6 s. Ignore the algorithm iteration time, software system processing time, and other preparation time, and only calculate the time of the pattern transfer process. After three verification experiments for each algorithm (including DODPACO), the average time of each algorithm is shown in Table 5.

#### 3.2.3. Discussion

The optimal convergence results of EDHACO, PSO-ACO-3opt, ACS, MMAS, and DODPACO 5 algorithms are shown in Figure 7. Figure 7 shows that DODPACO has the fastest convergence speed, and its solution is also optimal, with a value of 677 mm. Figure 8 is a path planning diagram when the shortest path value of the DODPACO algorithm is 677 mm. For this planning diagram, the initial (end) position coordinate of the 3D mobile platform is (1, 4). Figure 9 is a visual comparison diagram of the optimal value of the simulation path and the average time spent on pattern transfer. In the simulation, it is not difficult to find out from Figure 9 and Table 5 that the DODPACO algorithm for this problem has an optimal path of 677 mm. The path of the DODPACO algorithm is 45.3% shorter than that of no path planning, which is about 1% shorter than the suboptimal algorithm. In the verification experiment, the DODPACO algorithm saves 34.3% more time compared with no path planning, which is about 1% shorter than the suboptimal algorithm. From the simulation results and verification experiments, we can conclude that the DODPACO algorithm has the fastest convergence speed and the best solution quality, which is suitable for solving large-scale problems.

## 4. Conclusions

In this paper, a new dual-operator and dual-population ant colony algorithm (DODPACO) was proposed for ant colony algorithm to solve the problems of local optimal and slow convergence when solving large-scale TSP problems. The entropy automatically divided the number of ant colonies, the white ant colony optimized the convergence speed, and the yellow ant colony optimized the solution. The key data of simulation and verification experiments were: the error rate of the DODPACO algorithm was between 0 and 0.92, and the maximum error rate of the ACS algorithm and the MMAS algorithm was 6.58; the average time spent showed that the DODPACO algorithm proposed here had the shortest average time, which saved 34.3% of time compared with no path planning, which was about 1% shorter than the suboptimal algorithm. The importance of this article can be summarized as follows: (1) The simulation results showed that the DODPACO algorithm was superior in solving large-scale problems in terms of the solution and the convergence speed. (2) A path planning model was established to apply in path planning for DMD pattern transfer and other digital mask pattern transfer. (3) In the verification experiments, this algorithm effectively improved the DMD pattern transfer efficiency, providing a good case for the digital mask production of PCB and playing a positive role in the efficient application and promotion of digital mask lithography of DMD. In the future work, we will further study the updating model of adaptive pheromone of yellow white ant colony algorithm, reduce the value of standard deviation, and improve the robustness of the algorithm. The DODPACO algorithm will be widely applied in other large-area lithography exposure fields, such as MEMS production, micro-optical device processing, 3D micro-nano structure, integrated circuit production, etc.

## Figures and Tables

**Figure 1 entropy-22-00295-f001:**
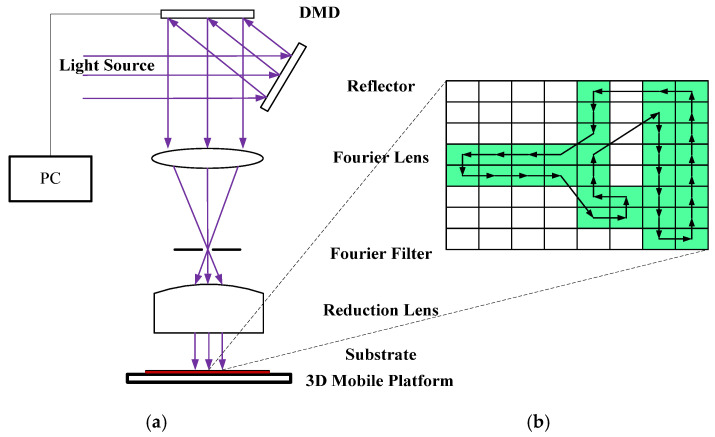
(**a**) Schematic diagram of a DMD projection pattern transfer lithography system, (**b**) an enlarged image of a substrate by path planning after pattern transfer.

**Figure 2 entropy-22-00295-f002:**
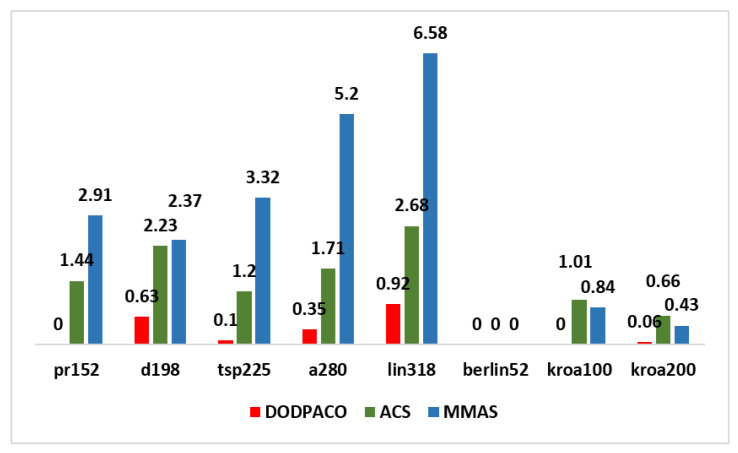
Comparison of error rates of three algorithms for eight groups of data simulation.

**Figure 3 entropy-22-00295-f003:**
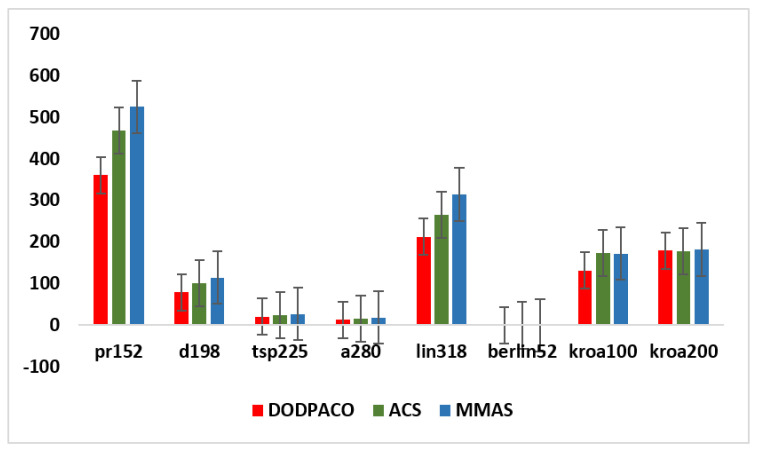
Comparison of standard deviation of three algorithms for eight groups of data simulation.

**Figure 4 entropy-22-00295-f004:**
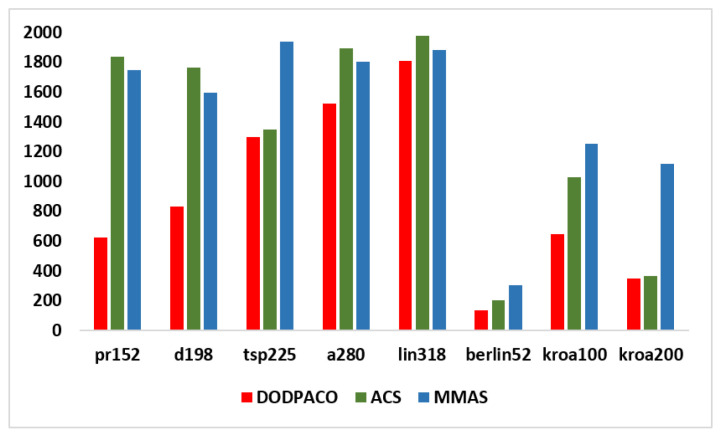
Comparison of minimum iterations of three algorithms for eight groups of data simulation.

**Figure 5 entropy-22-00295-f005:**
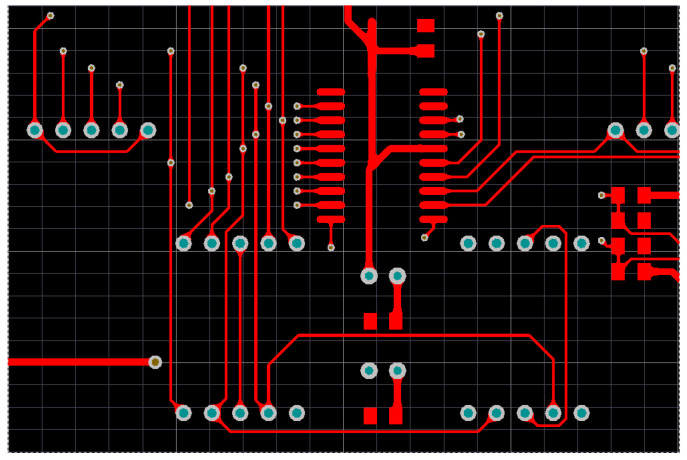
Screenshot of the top layer of a PCB.

**Figure 6 entropy-22-00295-f006:**
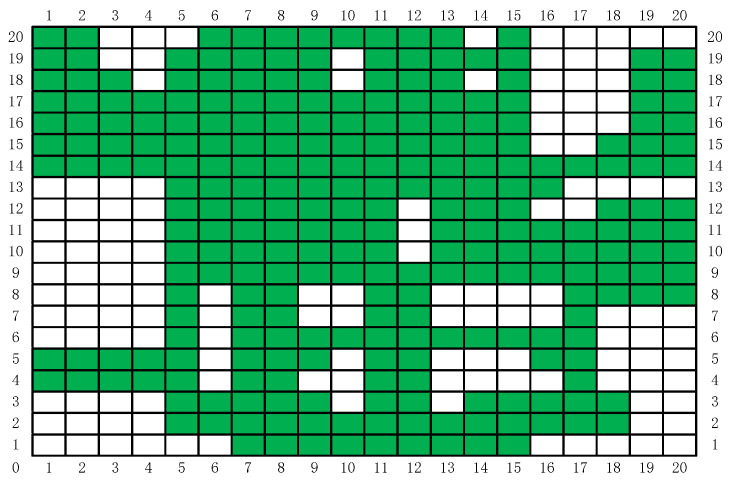
Marking diagram with PCB circuit in Figure 5.

**Figure 7 entropy-22-00295-f007:**
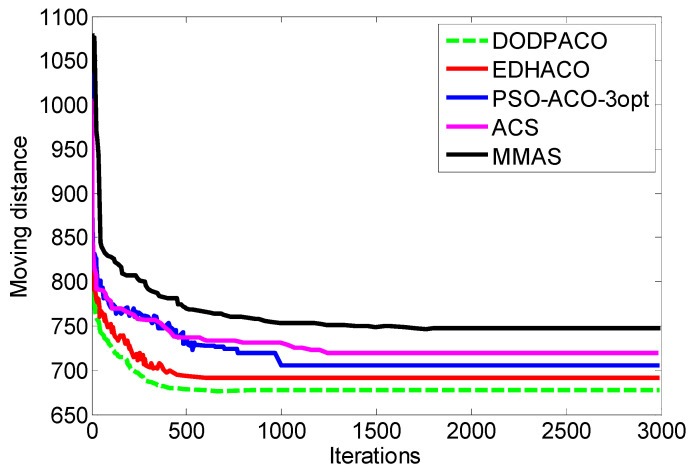
Simulation diagram of algebra and the optimal solution.

**Figure 8 entropy-22-00295-f008:**
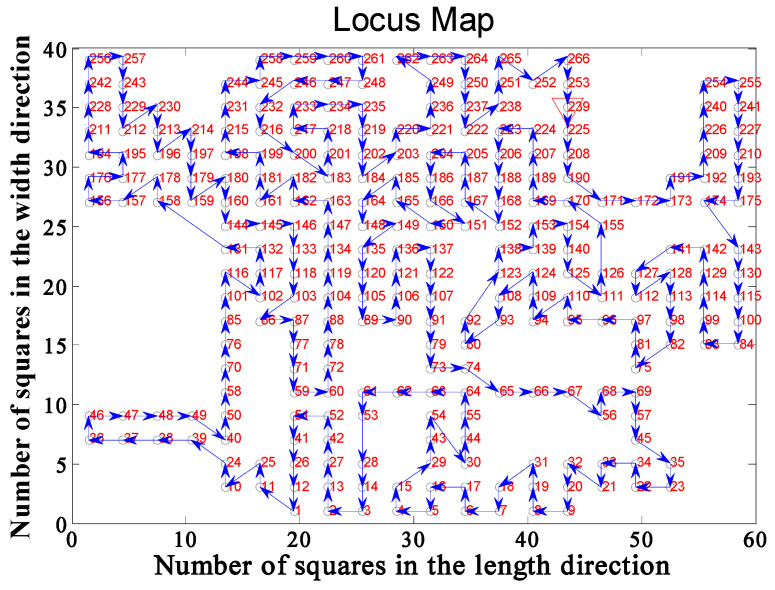
Schematic diagram of scanning path after planning.

**Figure 9 entropy-22-00295-f009:**
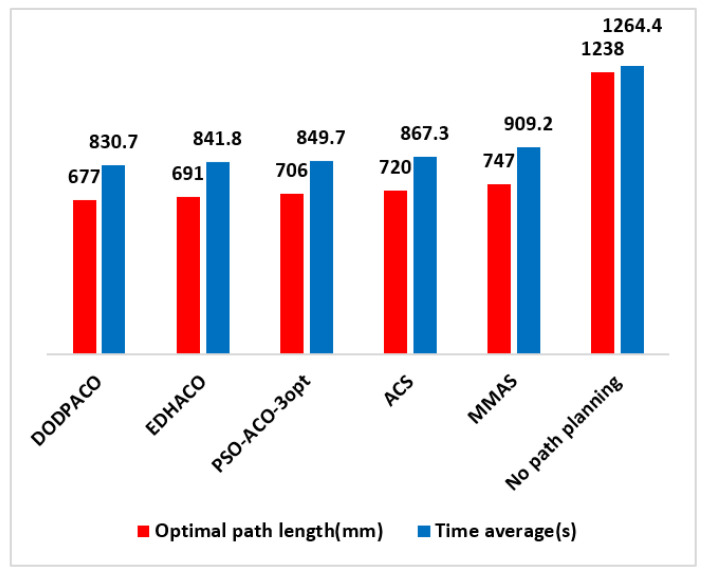
Optimal length of simulation path planning and average time of three graph transitions for the verification example.

**Table 1 entropy-22-00295-t001:** Parameter setting table used in the algorithm.

α	β	k	θ	ε	$\lambda_{1}$	$\lambda_{2}$	ρ
1	5	8	0.80	0.70	3/n	6/n	0.28

**Table 2 entropy-22-00295-t002:** Initialization value of ant colony number in the city test sets.

pr152	d198	TSP225	a280	lin318	berlin52	kroa100	kroa200
95	140	155	175	220	126	134	182

**Table 3 entropy-22-00295-t003:** Performance comparison of DODPACO, ACS, and MMAS in different TSP instances.

Instance	Opt	Algorithms	Best	Mean	Error rate	Standard Deviation	Convergence
pr152	73,682	DODPACO	**73,683**	**73,905**	**0.00**	**360**	**624**
		ACS	74,742	74,929	1.44	467	1838
		MMAS	75,829	76,056	2.91	524	1745
d198	15,780	DODPACO	**15,790**	**15,896**	**0.63**	**79**	**832**
		ACS	16,132	16,172	2.23	101	1765
		MMAS	16,154	16,20	2.37	114	1596
TSP225	3916	DODPACO	**3920**	**3963**	**0.10**	**21**	**1298**
		ACS	3963	3973	1.20	25	1349
		MMAS	4046	4058	3.32	27	1940
a280	2579	DODPACO	**2588**	**2591**	**0.35**	**13**	**1521**
		ACS	2623	2630	1.71	16	1891
		MMAS	2713	2721	5.20	19	1805
lin318	42,029	DODPACO	**42,416**	**42,458.42**	**0.92**	**212**	**1806**
		ACS	43,155	43,263	2.68	265	1979
		MMAS	44,794	44,928	6.58	314	1881
berlin52	7542	DODPACO	**7542**	**7542**	**0**	**0**	**134**
		ACS	7542	7542	0	0	200
		MMAS	7542	7542	0	0	304
kroa100	26,524	DODPACO	**21,282**	**21,286**	**0**	**132**	**645**
		ACS	26,793	26,938	1.01	174	1029
		MMAS	26,746	26,562	0.84	172	1254
kroa200	29,368	DODPACO	**29,387**	**29,506**	**0.06**	**179**	**349**
		ACS	29,561	30,732	0.66	177	367
		MMAS	29,495	30,435	0.43	182	1120

**Table 4 entropy-22-00295-t004:** The computational results of the proposed method and other methods in the literature.

Algorithms	Instance	pr152	d198	TSP225	a280	lin318	berlin52	kroa100	kroa200
	Known Best Solution	73,682	15,780	3916	2579	42,029	7542	21,282	29,368
PCCACO	best	/	15,814	3937	/	42,461	7542	21,282	29,391
	mean	/	16,463	3981	/	42,933	7542	21,383	29,485
EDHACO	best	**73,682**	/	/	/	43,291	/	21,282	29,694
	mean	74,251.6	/	/	/	43,926.3	/	21,355.13	30,391
ICMPACO	best	/	/	4106	/	/	7548.6	/	31,267
	mean	/	/	4214	/	/	7621.36	/	32,086
PSO-ACO-3opt	best	/	/	4135	/	/	7542	21,301	29,468
	mean	/	/	4250	/	/	7543.2	21,445.1	29,957
HHACO	best	/	/	3998	/	/	/	/	/
	mean	/	/	4113	/	/	/	/	/
CCMACO	best	/	/	3926	2592	42,475	/	21,282	29,399
	mean	/	/	4086.5	2682.6	42,682.7	/	21,488.3	29,834.8
Proposed Method DODPACO	best	73,683	**15,790**	**3920**	**2588**	**42,416**	**7542**	**21,282**	**29,387**
	mean	**73,905**	**15,896**	**3963**	**2591**	**42,458**	**7542**	**21,286**	**29,** **356**

**Table 5 entropy-22-00295-t005:** Path optimization value and average time of graph transfer in verification experiments.

Algorithms	DODPACO	EDHACO	PSO-ACO-3opt	ACS	MMAS	No Path Planning
Optimal path length (mm)	677	691	706	720	747	1238
Time average (s)	830.7	841.8	849.7	867.3	909.2	1264.4
Time savings compared with no-path planning (%)	34.3	33.4	32.8	31.4	28.1	0

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
