# Peer review of "Path Planning of Pattern Transfer Based on Dual-Operator and a Dual-Population Ant Colony Algorithm for Digital Mask Projection Lithography"

_entropy, 2020, doi:10.3390/e22030295_

Round 1
Reviewer 1 Report
"Path Planning of Pattern Transfer Based on Dual-operator and Dual-population Ant Colony Algorithm for Digital Mask Projection Lithography"
The Authors present and discuss results on a new dual operator and dual population ant colony (DODPACO) algorithm to achieve the goals of accelerating 21 convergence and improving the quality of the solution. The manuscript is well-written, however a few aspects need further and thorough revision. The Authors need to also take into account the "General Considerations" of the "Manuscript Preparation", available at https://www.mdpi.com/journal/entropy/instructions.
- Abstract: meaningful, but please rephrase it to provide more on the actual and significant values / data;
- Graphical abstract: not provided;
- Highlights: not provided;
- Keywords: meaningful.
1. Introduction
- this section is comprehensive, yet the Authors use long phrases and in some cases it's rather difficult to follow; please rephrase it.
-
the Authors must further insist on the importance and novelty of their work with respect to literature;
- further explain on your choice on this approach; the "Related Work" section would be more appropriate here, not as a separate section.
2. Experimental approach / Materials and methods
- this section will include "3. Algorithm Principle and Algorithm Improvement" and any discussion with respect to the "Algorithm Flow"; therefore, the Authors need to thoroughly rephrase this section.
3. Results and discussion
- as a general overview / remark to this section: it's not sufficiently presented and discussed, the Authors need to further discuss their results in a more correlated manner;
- this section will include "4. Simulation and Experiment" and "5. Experiments";
- to conclude, the Authors need to thoroughly rephrase it.
4. Conclusion
-
this section is meaningful, but please rephrase it to insist more on the novelty and importance of your work; provide more on the actual and significant values / data.
Reviewer 2 Report
This paper reports a novel heuristic algorithm for path planning in digital mask projection lithography. The authors developed a new dual operator and dual population ant colony (DODPACO) algorithm that can solve the traveling salesman problem more efficiently than the traditional genetic algorithm and ant colony optimization. This algorithm showed better optimization consistency and achieved shorter lithography steps compared to a range of existing algorithms. The major problem is that the writing is wordy and confusing. Therefore, I strongly support the acceptance of the manuscript if they can address the comments and problems including:
- Entropy is a journal about information theory, the majority of audiences are not familiar with photolithography in general. Therefore, I would suggest adding a large overview scheme that demonstrates:
- The process of digital-mask lithography in general
- The a zoomed in image (scheme) of the lithography path similar to the current Figure 7
- The path optimization challenge and its connection to the traveling salesman problem
- The general strategy of the ant colony optimization
- For the introduction, line 41 introduced the concept of digital mask lithography. But what is a digital mask and how does it work? How is it different from the conventional contact mask lithography methods and why is it better? Summarize in 1-2 sentences.
- On line 63, why is PCB important? It sounds like from line 48 that PCB is just one example that uses DMD.
- On line 66, why is DODPACO selected instead of the genetic algorithm. I can guess from the later description that methods based on ant-colony optimization are natural approaches to the traveling salesman problem in the path planning. I believe that these points should be stated clearly before line 66.
- Lines 70-94 should be more concise and placed before the “in this paper” summary on line 64.
- Section 2 “Related Work” is not necessary. I don’t think Entropy requires this section either.
- Is section 3.1 describing only the ACS algorithm published before? Then this section should be shrunk by at least 50%. Just cite the publications about ACS here.
- The overall writing needs to be improved significantly. Issues including but not limited to:
- When listing items, use parallel structures. For example, lines 39-40: “high cost” and “time-consuming” are adjectives while “lack of flexibility” is a noun. Also the “of” on line 40 can only follow nouns, but not adjectives.
- Line 42, the “such as” follows the noun “problem”, but “electron beam …. FIB” are not example of “problems”. What does “such as” refer to.
- I don’t understand the sentence on lines 49-52.
- The scalability issue described on lines 54-55 “The larger the substrate area is, the larger the number of steps and the more time it takes” is intuitive and can probably be assumed. No need to describe it.
- Line 61, “which” should refer to the “transfer pattern” as written. Probably needs to be corrected.
Round 2
Reviewer 1 Report
The Authors correctly addressed the issues raised during the review process.
The manuscript is now suitable for publication in the MDPI journal "Entropy".